# Full-Length Transcriptome Sequencing Reveals Treg-Specific Isoform Expression upon Activation

**DOI:** 10.3390/ijms26136302

**Published:** 2025-06-30

**Authors:** Yohei Sato, Erika Osada, Yoshinobu Manome

**Affiliations:** 1Project Research Unit, Laboratory of Immune Cell Therapy, The Jikei University School of Medicine, Tokyo 105-8461, Japan; 2Core Research Facilities, Research Center for Medical Sciences, The Jikei University School of Medicine, Tokyo 105-8461, Japan; osada411.m45-410@jikei.ac.jp (E.O.); manome@jikei.ac.jp (Y.M.)

**Keywords:** regulatory T cells (Tregs), FOXP3, RNA-seq, Iso-seq, PD-L1

## Abstract

FOXP3+ regulatory T cells (Tregs) play a central role in the regulation of the immune system. Human Tregs preferentially express a FOXP3 isoform known as delta 2, which lacks exon 2. In addition to FOXP3, Tregs also express isoforms of other Treg-related molecules, such as CTLA-4 and IKZF-2. It is hypothesized that Tregs possess a unique isoform repertoire based on their unique gene and isoform expression profiles, which include FOXP3. Here, we identified a Treg-specific unique isoform repertoire confirmed by long-read high-throughput isoform sequencing called Iso-seq, which is uniquely capable of providing data on genome-wide isoform usage. Notably, while conventional T cells (Tconvs) do not exhibit this pattern, Tregs preferentially express the full-length FOXP3 isoform. Interestingly, the preferential expression of ICOS and PD-L1 upon T-cell receptor (TCR) stimulation was noted in activated Tregs but not in Tconvs or non-activated Tregs. Moreover, using a PD-L1 antibody blockade on Tregs did not diminish FOXP3 expression; however, it significantly reduced the suppressive function. Therefore, Tregs may have a unique isoform repertoire, which becomes pronounced upon polyclonal TCR stimulation.

## 1. Introduction

CD4^+^ T cells are essential immune cells that regulate the adaptive immune response to pathogens. While conventional T cells (Tconvs) recognize foreign antigens through antigen-presenting cells via MHC molecules, regulatory T cells (Tregs) are a unique subset of CD4^+^ T cells that play a central role in maintaining self-tolerance by recognizing self-antigens [1]. FOXP3 serves as a master transcription factor that governs the gene expression of Tregs [2,3]. Tregs are identified based on the CD4^+^CD25^+^CD127^low^ population and are characterized by a high and stable FOXP3 expression [3,4]. In addition to surface molecule expressions, Tregs are distinguished by their unique gene expression profile [5]. Transcriptome analysis has been conducted to characterize Tregs using various modalities, including microarray, bulk RNA sequencing (bulk RNA-seq), and single-cell RNA sequencing (single cell RNA-seq). Tregs have a unique gene expression profile, suggesting that they are an independent population compared to Tconvs. This is most likely due to their high and stable expression of FOXP3, which results from thymic differentiation driven by the expression of TCR-specific self-antigens. This process directly promotes the expression of IKZF-2 (Helios). Binding targets of FOXP3, including IKZF-2, were identified using ChIP-seq [6]. Consequently, the cellular and molecular identities of Tregs have been successfully identified using transcriptome, proteome, and epigenome analyses.

RNA-seq is a unique modality that allows the investigation of genome-wide gene expression profiles measured through short-read sequencing. Transcriptome analysis of the immune cells has been conducted using either bulk or single-cell RNA-seq. Tregs possess a distinct molecular identity, characterized by FOXP3 expression, and Treg-related gene signatures are maintained even at the single-cell level [5,7,8]. Hence, Tregs can be identified using single-cell RNA-seq based on their unique gene expression phenotypes. Indeed, single-cell RNA-seq has successfully revealed the development and tissue adaptation of Tregs [9]. These Treg-specific gene expression profiles shed light on Treg biology and enhance the clinical application of Tregs. Therefore, Treg biology is currently being investigated using multiple molecular approaches spanning from protein to RNA.

Alternative splicing is another mechanism that generates transcripts leading to significant changes in gene expression. Indeed, cancer cells utilize isoform expression or their “isoform repertoire” for survival and escape from immune cells [10]. Immune cells are traditionally known to express specific isoforms according to cell lineage [11]. CD4^+^ T cells have been reported to acquire isoform expression upon activation; however, characterization of genome-wide isoform expression profiles using conventional sequencing approaches has proven challenging. Tregs have two predominant FOXP3 isoforms, the full-length isoform and the delta 2 isoform, which lacks exon 2 [12]. IKZF-2, another transcription factor expressed in Tregs, is also known to produce an isoform via alternative splicing [13]. However, due to technical limitations, it was not possible to characterize the isoform expression profile.

T-cell activation is a critical factor that can potentially influence gene expression. Tregs become functional after activation via T-cell receptor (TCR) stimulation [14]. The alternative splicing of CD4^+^ T cells is also influenced by activation [15]. Therefore, it is hypothesized that activated Tregs may have a unique isoform expression following TCR stimulation. It is not readily possible to study genome-wide isoform expression profiles, especially for a rare population of immune cells such as Tregs, owing to the limitation of number and phenotypic stability during expansion. To characterize genome-wide isoform expression profiles, Tregs were analyzed using long-read RNA-seq, also known as Iso-seq. Furthermore, we compared isoform expression profiles between Tconvs and Tregs with and without TCR stimulation, which could potentially influence both gene and isoform expression. Here, we demonstrate that Treg-specific isoform expression can be identified by conventional RNA-seq and Iso-seq analyses. PD-L1 and ICOS were significantly upregulated in activated Tregs compared with Tconvs, especially after TCR stimulation. Interestingly, PD-L1 was prominently upregulated in Tregs compared to that in non-activated Tregs and Tconvs. Additionally, the antibody-mediated blocking of PD-L1 molecules on Tregs inhibited the suppressive function, suggesting that PD-L1 is critical for the regulatory function of Tregs.

## 2. Results

### 2.1. Tregs Showed Unique Gene Expression Profiles Compared with Tconvs

Tregs and Tconvs isolated from frozen PBMCs (n = 3, healthy donors) were characterized and sorted using FACS (Figure 1a). The analysis showed that the expressions of FOXP3, CD25, and CD127 differed significantly between Treg and Tconv populations but were homogeneous among the three independent PBMC donors used in this study (Appendix A). RNA-seq analysis demonstrated that Tregs maintained a unique gene expression profile compared with the Tconvs isolated from the same PBMC donors (Figure 1b). Based on conventional RNA-seq analysis, 1004 differentially expressed genes (DEGs) were identified between Tconvs and Tregs isolated from the same PBMC donors (Figure 1c). As expected, FOXP3 expression was significantly higher in Tregs than in Tconvs (*p* = 4.73 × 10^−159^). In addition, known Treg-related molecules, including IL2RA (*p* = 2.58 × 10^−60^), RTKN2 (*p* = 6.52 × 10^−50^), IKZF-2 (*p* = 6.23 × 10^−43^), TNFRSF9 (*p* = 5.57 × 10^−42^), and IL2RB (*p* = 3.98 × 10^−36^) were significantly enriched in Tregs compared with Tconvs. Gene set enrichment analysis (GSEA) was used to confirm the Treg-related gene signatures in Tregs in comparison with Tconvs (Figure 1d, Appendix A). The data confirmed that Tregs have a unique gene expression profile compared with that of Tconvs.

### 2.2. Iso-Seq Identified Treg-Specific Isoform Expression upon Activation

Conventional short-read RNA-seq is a robust methodology for investigating genome-wide gene expression in both bulk- and single-cell populations. Conventional bulk or single-cell RNA-seq identifies gene expression based on short-read sequencing data. Furthermore, it is technically possible to distinguish different isoform usages even from conventional short-read RNA-seq data through computational calculations such as MAJIQ. In terms of accuracy, quantification of isoform usage based on long-read RNA-seq analysis methods, including CAGE-seq and Iso-seq, is preferred to conventional short-read RNA-seq. Iso-seq identified unique isoform usage in both non-activated and activated Tregs compared to their Tconv counterparts (Figure 2a,b). Activated cells acquire unique gene expression patterns.

Several genes express multiple isoforms in various immune cells. In CD4^+^ T cells, CD44, CD45, and MALT1 are known to express multiple isoforms. Our data are compatible with previous literature, showing a unique isoform expression profile in CD4^+^ T cells (Appendix A). To further characterize Treg-specific gene expression, we investigated Treg-related gene expression profiles using Iso-seq. Interestingly, activated Tregs showed unique isoform usage compared to activated Tconvs with regard to genes related to FOXP3, such as IKZF-2, CTLA-4, GITR, TIGIT, and ICOS (Figure 2c). The upregulation of Treg-related molecules (FOXP3, IKZF-2, CTLA-4, GITR, TIGIT, and ICOS) was further validated using qPCR (Figure 3c). Collectively, Treg-related molecules were upregulated in terms of total gene expression and isoform variant expression. This suggests that activated Tregs exhibit unique gene expression profiles in both quality and quantity.

### 2.3. Tregs, Not Tconvs, Preferentially Expressed Full Length FOXP3 Isoform

Human Tregs express two major FOXP3 isoforms, known as full-length and delta 2 isoforms. The full-length FOXP3 isoform may play an essential role in Treg development and differentiation. This idea is supported by the observation that a patient with a specific mutation in exon 2 only expressed the full-length isoform [16]. Although Tregs have two different isoforms, it is technically difficult to distinguish between them because of their similarity, which makes it difficult to design isoform-specific primers to distinguish them by quantitative PCR.

Iso-seq analysis revealed that upon activation, Tregs preferentially express the full-length FOXP3 isoform, whereas Tconvs expressed both isoforms almost equally (Figure 2a). It is also important for Tregs to express the delta 2 isoform upon activation. Moreover, it was possible to identify different isoforms of activated Tregs with regard to several Treg-related molecules known to express these isoforms. In addition to the molecular approaches, we further confirmed the upregulation of Treg-related molecules using FACS (Figure 3a,b). Upregulation of Treg-related molecules, including FOXP3 (*p* < 0.0001), GITR (*p* < 0.001), ICOS (*p* < 0.001), CTLA-4 (*p* < 0.01), TIGIT (*p* < 0.01), and IKZF-2 (*p* < 0.05), was further validated by FACS (Figure 3d). Furthermore, this upregulation was maintained after polyclonal CD3/28 stimulation. Notably, among these molecules, only ICOS was significantly upregulated in activated Tregs compared with resting Tregs (*p* < 0.01). Significant upregulation of ICOS was observed in different analyses (Iso-seq, qPCR, and FACS). Therefore, ICOS is most strongly correlated with activated Tregs.

### 2.4. Activated Tregs, Not Resting Tregs, Preferentially Expressed CD80 and PD-L1

Interestingly, Iso-seq analysis revealed that activated Tregs preferentially expressed CD80 and PD-L1 compared with activated Tconvs and unstimulated Tregs (Figure 4a). However, significant upregulation of CD86 and PD-1 in Tregs could not be detected, regardless of analytical method. Additionally, qPCR analysis in activated and non-activated Tregs identified a significant upregulation of PD-L1 in Tregs upon activation (*p* < 0.05); however, the same was not true for CD80 (*p* = 0.30), CD86 (*p* = 0.19), or PD-1 (*p* = 0.15) (Figure 4b). It is not likely that PD-L1 expression was due to the contamination of the antigen-presenting cells, since other major monocyte/macrophage markers, including CD14 and CD16, were not identified either by RNA-seq or Iso-seq (Appendix A). Furthermore, RNA-seq and qPCR confirmed that PD-L1 expression was significantly upregulated in Tregs upon activation. Collectively, these results suggest that PD-L1 expression is associated with activated Tregs.

In addition to the molecular approaches, we further confirmed the upregulation of surface molecule expression on activated Tregs using FACS (Figure 5a,b). Compared to the activated Tconvs, CD80 (*p* < 0.05), CD86 (*p* < 0.05), PD-1 (*p* < 0.01), and PD-L1 (*p* < 0.001) were significantly upregulated in the activated Tregs (Figure 5c). Notably, PD-L1 was more significantly upregulated in activated Tregs than in non-activated Tregs (*p* < 0.0001), whereas the other molecules were equally expressed in both activated and non-activated Tregs. Based on these findings, we hypothesized that PD-L1 upregulation in activated Tregs may contribute to stable FOXP3 expression and suppressive function.

### 2.5. Inhibition of PD-L1 Expression on Tregs Resulted in the Loss of Suppressive Function

According to the Iso-seq analysis, it is hypothesized that PD-L1 contributes to the stabilization of FOXP3 and its suppressive function in human Tregs. To investigate the importance of PD-L1 expression on Tregs, FOXP3 expression and suppressive function were assessed after antibody blocking of surface molecules, including PD-1, PD-L1, CD80, and CD86. Antibody blocking did not diminish FOXP3 expression on Tregs (Appendix A). It was not possible to observe a significant difference at the PD-L1 antibody concentration of 0.1 μg/mL across the different ratios of Tregs versus Tconvs (Figure 6a). Interestingly, inhibition of PD-L1 on Tregs resulted in the loss of suppressive function at a PD-L1 antibody concentration of 1.0 (*p* < 0.01)–10 μg/mL (*p* < 0.05), especially when Tregs were co-cultured at a 1:1 or 2:1 ratio together with Tconvs. This is in contrast to the effect noted in the isotype control (Figure 6b,c). In contrast, PD-1, CD80, and CD86 antibody inhibition of Tregs neither enhanced nor diminished suppressive function among different antibody concentrations (Figure 6a–c). Collectively, these findings indicate that PD-L1 expression on activated Tregs is important for the maintenance of FOXP3 expression and its suppressive function.

## 3. Discussion

Tregs represent a unique subset of immune cells that regulate self-tolerance during the adaptive immune response [17]. FOXP3+ Tregs exhibit a unique FOXP3 isoform expression [12]. The non-overlapping roles of FOXP3 isoforms have been elucidated through the integration of CRISPR/*Cas9* gene editing and the viral gene transfer approach [18]. However, it remained unclear whether Tregs had a genome-wide unique isoform expression—the so-called “isoform repertoire.” Iso-seq is a recently developed long-read sequencing technology that enables genome-wide isoform identification. Several types of cancer cells are known to have unique isoform repertoires, based on iso-seq analysis. It has already been identified that immune cells have different isoform usages upon activation; however, the isoform repertoire of Tregs based on long-read sequencing has not been reported previously. In fact, in addition to FOXP3, other representative Treg molecules such as CTLA-4 and IKZF-2 express isoforms, especially within Tregs. Therefore, it is essential to identify the genome-wide isoform repertoire of Tregs using long-read RNA-seq.

Treg-related molecules are expressed by FOXP3+ Tregs and play a critical role in their suppressive function. These molecules have long been investigated as potential therapeutic targets for cancer immunotherapy, with some already transitioning into clinical therapeutic options. PD-1 and CTLA-4 are co-inhibitory molecules that are highly expressed by Tregs [19] and have been extensively studied to identify novel targets for cancer immunotherapy. Clinical practice has already incorporated PD-1, PD-L1, CTLA-4, and LAG-3 blockades. Other molecules such as TIGIT, ICOS, and GITR are currently under investigation as alternative immunotherapy targets. We confirmed the upregulation of these molecules by RNA-seq, Iso-seq, FACS, and qPCR. Notably, ICOS and PD-L1 were significantly upregulated in Tregs upon activation, whereas others, despite being highly expressed in Tregs, were not significantly influenced by activation. Therefore, ICOS and PD-L1 may serve as more effective targets against activated Tregs, which elicit a more aggressive tumor-inhibitory immune response in the tumor microenvironment than non-activated Tregs.

Recently, clinical observations have demonstrated that Tregs were shown to expand and become suppressive after PD-1 blockade [20]. PD-L1, usually expressed by antigen-presenting cells, is unexpectedly upregulated in Tregs [21]. Furthermore, the interaction between PD-1 and PD-L1 is critical for the Treg development and its suppressive function. Similar to PD-1, CTLA-4 blockade has been reported to expand Tregs. Collectively, the mechanism of checkpoint blockade has primarily been attributed to the enhancement of Tconv proliferation and antitumor effects. However, checkpoint blockade may not inhibit Treg proliferation and impair suppressive function. Our data support the fact that checkpoint blockade does not directly dampen FOXP3 expression or the suppressive function of Tregs. However, in our study, it was demonstrated that only PD-L1 blockade might inhibit suppressive function, especially when Tregs were co-cultured with almost the same number of Tconvs. Nevertheless, CD80/CD86 blockade also partially inhibits T cell proliferation in limited experimental conditions. Overall, Tregs maintain suppressive function under the checkpoint blockade except PD-L1 blockade, which might potentially inhibit activated Treg function independent of FOXP3 expression. Currently, a bispecific antibody targeting both PD-L1 and ICOS has been developed as a novel immune checkpoint blockade [22]. Based on our findings, bispecific PD-L1 and ICOS inhibition will efficiently inhibit activated Tregs, which may contribute to the immune escape of cancer cells to evade host immune surveillance.

CD80 and CD86 are expressed by activated CD4^+^ T cells with CD80 contributing to the stabilization of FOXP3 expression [23]. Our previous RNA-seq analysis identified significant upregulation of CD80 in freshly isolated Tregs compared to that in Tconvs [24]. Tregs have also been shown to express CD80 and CD86; however, the function of these molecules on Tregs has not been fully disclosed [25]. Our findings do not contradict a previous description that the upregulation of CD80 and CD86 occurred with PD-L1 upregulation. Traditionally, CD80, CD86, and PD-L1 were considered surface molecules preferentially expressed by antigen-presenting cells, rather than T cells. However, they are now known to be expressed by Tregs upon activation [26]. This indicated that Tregs express common antigen-presenting cell molecules upon activation. These surface molecules are ligands for PD-1 and CTLA-4; therefore, they might be associated with Treg–Treg interactions or other forms of intercellular communication among immune cells.

In this study, we identified Treg-specific isoform usage upon activation. Whereas other Treg-related molecules TIGIT, GITR, PD-1, and CTLA-4 are upregulated independent of activation, ICOS and PD-L1 were most significantly upregulated upon activation compared to that in non-activated Tregs. Due to the limited sample number, we were not able to perform qPCR for multiple donors (n > 10) to consolidate our findings from RNA-seq, Iso-seq, and Flow cytometry. Further studies are warranted to consolidate isoform expression with regards to each key gene related to Treg function, including FOXP3, IKZF-2, and PD-L1. Furthermore, PD-L1 antibody blockade inhibits this suppressive function, suggesting that PD-L1 contributes to suppression. In addition to PD-L1, Tregs preferentially express CD80 and CD86, both of which are usually expressed by antigen-presenting cells but not by CD4^+^ T cells. Collectively, these findings indicate that Tregs have unique gene and isoform expressions upon activation, and these molecules can potentially be used as targets for cancer immunotherapy and immunomodulation in autoimmunity.

## 4. Materials and Methods

### 4.1. Human Peripheral Blood Mononuclear Cell (PBMC) Thawing

Frozen human PBMCs isolated from healthy donors were purchased from Precision Medicine (n = 4, for bulk RNA-seq) and Lonza (n = 4, for other studies) and then stored in liquid nitrogen. Frozen PBMCs were thawed in 10 mL of PBS (Gibco, 10-010-023, Thermo Fisher Scientific, Waltham, MA, USA) supplemented with 25 U/mL of benzonase nuclease (Sigma Aldrich, E1014, St. Louis, MO, USA). Cells were spun down at 400 g for 5 min at 4 °C. After centrifugation, cell count and viability were measured using the Invitrogen Countess II Automated Cell Counter following trypan blue staining (Thermo Fisher Scientific, 15250061). PBMCs with reduced viability (Live cell percentage < 80%) were excluded from further study. CD4^+^ T cells, including Tregs isolated by the protocol described in the following section, were cultured in X-vivo 15 media supplemented with gentamicin (Lonza, 02-053Q, Basel, Switzerland) and 5% human serum from pooled male AB plasma (Sigma-Aldrich, BP2525100) in the presence of 100 U/mL of recombinant human (rh) IL-2 (Peprotech, 200-02-50 ug, Cranbury, NJ, USA) for 24 h at 37 °C and harvested for the analysis.

### 4.2. Treg Isolation and Activation

Tregs were isolated using the EasySep Human CD4^+^CD127^low^CD25^+^ Treg Isolation Kit (Stem Cell Technologies, ST-18063, Vancouver, BC, Canada) according to the manufacture’s protocol. Tregs (CD4^+^CD127^low^CD25^+^ cells) and Tconvs (CD4^+^CD25^−^ cells) were obtained from the same PBMC donors. The quality of Tregs isolated from PBMC was assessed by fluorescence activated single cell sorting (FACS). Freshly isolated Tregs and Tconvs were activated by polyclonal TCR stimulation with a plate-bound CD3/CD28 antibody (BioLegend, 317301 and 302901, San Diego, CA, USA). The CD4^+^CD25^−^ fraction was used as an internal control for Tregs.

### 4.3. Fluorescence-Activated Single Cell Sorting (FACS)

CD4^+^ T cells, including both freshly isolated and stimulated Tconvs/Tregs, were resuspended in FACS Buffer (PBS supplemented with 0.5% BSA and 2 mM EDTA) and stained with an antibody cocktail for 30 min. After surface staining, intracellular staining for FOXP3 was performed using the *Foxp3*/Transcription Factor Staining Buffer set (eBioscience, 00-5523-00, San Diego, CA, USA). The antibodies used for FACS analysis are listed in Appendix A. Data were acquired using a MACSQuant Analyzer (Miltenyi, Waltham, MA, USA) or FACSAria III (BD Biosciences, Friendswood, TX, USA) and analyzed using FlowJo 10.7.1 software (FlowJo LLC, Ashland, OR, USA).

### 4.4. RNA Extraction

Total RNA was extracted from the Tconvs and Tregs using an RNAeasy Micro Kit (QIAGEN, 74004, Hilden, Germany). RNA quality was measured using a Nanodrop (Thermo Fisher Scientific) and a bioanalyzer (Agilent Technologies, Santa Clara, CA, USA). Samples with an A_260/280_ ratio > 2.0 and/or RIN > 7.5 were set as cut-off values for further analysis.

### 4.5. Quantitative Polymerase Chain Reaction (qPCR)

The cDNA was synthesized from 100 ng of RNA using a PrimeScript RT Master Mix (Takara Bio, RR0037A, San Jose, CA, USA). Synthesized cDNA was amplified using a Thunderbird qPCR Probe Mix (Toyobo, Osaka, Japan, QPS-101) and a Taqman Gene Expression Assay (Applied Biosystems, Foster City, CA, USA) for 40 cycles. The cycle threshold (Ct) values of the target genes were measured and calculated using Quant Studio 5 (Thermo Fisher Scientific). Relative gene expression (hold changes) was normalized using the housekeeping gene (HPRT1) and calculated using the ΔCt method. A list of TaqMan gene expression assays is shown in Appendix A.

### 4.6. RNA-Seq

Total RNA was extracted from 30,000 Tregs and Tconvs and stored in QIAzol (QIAGEN, 79306). RNA quality was assessed using an Agilent 2100 Bioanalyzer (Agilent Technologies). RNA samples with RIN > 7.5 were used for RNA-seq library preparation. RNA-seq libraries were generated using a Nextera XT Library Prep Kit (Illumina, FC-131-1024, San Diego, CA, USA). Sequencing data were generated on an Illumina HiSeq 4000 (75 bp, paired-end). For RNA-seq analysis, FASTQ files obtained from RNA-seq were aligned into the human genome using STAR mapping software. Gene expression analysis was performed using R-studio software. Differential gene expression was analyzed using DEseq2 (adjusted *p*-values < 0.01), and a heat map was created using the Heatmap2 software. RNA-seq results are shown in Appendix A.

### 4.7. Long-Read RNA Sequencing (Iso-Seq)

Total RNA was extracted from 500,000 Tregs and Tconvs and stored in QIAzol (QIAGEN). RNA quality was assessed using an Agilent 2100 Bioanalyzer (Agilent Technologies). RNA samples with RIN > 7.5 were used for Iso-seq library preparation. The cDNA was synthesized using the NEB Next Single Cell/Low Input cDNA Synthesis & Amplification Module (New England Biolabs, E6420S, Ipswich, MA, USA). Iso-seq libraries were generated using the SMRTbell Express Template Prep Kit 2.0 (Pacific Bio, 100-938-900, Menlo Park, CA, USA). Sequencing data were generated using the PacBio Sequel II System (Pacific Bio). For Iso-seq analysis, HiFi reads obtained from the PacBio Sequel II System were aligned into the human genome, and isoform expression was counted using SQANTI3. Isoform expressions were shown by log_10_ transcripts per million (log_10_TPM). The Iso-seq results are shown in Appendix A.

### 4.8. Suppression Assay

The suppression assay for Tregs was performed as previously described. Briefly, responder CD4^+^CD25^−^ T cells (50,000 cells) labeled with 2 μM CFSE (Dojindo, C375, Rockville, MD, USA) and suppressor CD4^+^CD25^+^CD127^−^ cells (12,500–50,000 cells) labeled with 2 μM CytoRed (Dojindo, C400) were co-cultured at different concentrations (1:1–8:1) following activation by Dynabeads Human T-cell Activator CD3/CD28 (Gibco, DB11131) beads at a 1:25 bead-to-cell ratio. Suppressor CD4^+^CD25^+^CD127^−^ cells were incubated with anti-PD-1 (Biolegend, 329902), PD-L1 (Biolegend, 393602), CD80 (Biolegend, 305201), and CD86 (Biolegend, 374202) antibodies for 2–3 h at room temperature for blocking at the concentration of 0.1–10 μg/mL before being co-cultured with responder cells. After 96 h of stimulation, the proliferation of responder CD4^+^ T cells was counted using FACS.

### 4.9. Statistical Analyses

GraphPad Prism Software ver 9.0 (GraphPad software) was used for statistical analyses. All statistical analyses were performed using the two-tailed Student’s t-test or Mann–Whitney U-test. One-way ANOVA followed by Tukey’s multiple comparison tests were performed when comparing multiple conditions, and *p* values < 0.05 were considered as significant. Data are shown as mean ± SEM.

## Figures and Tables

**Figure 1 ijms-26-06302-f001:**
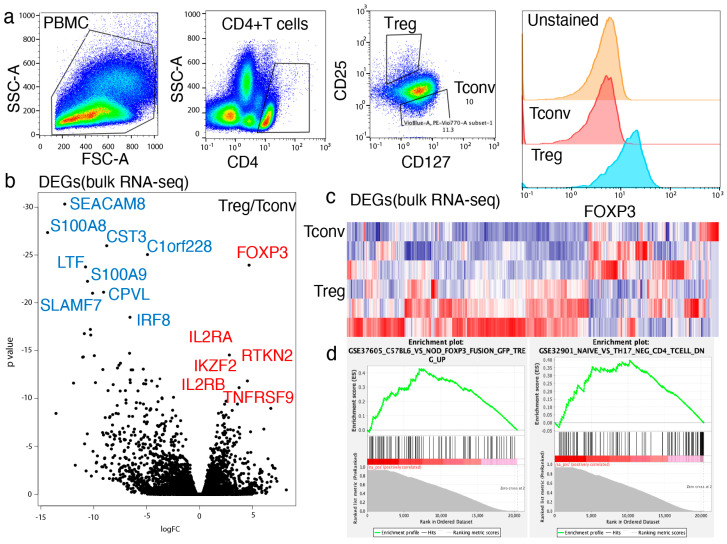
Tregs showed unique gene expression compared to Tconvs. (**a**) Tregs (CD4^+^CD25^+^CD127^low^) and Tconvs (CD4^+^CD25^−^) were identified and isolated by fluorescence activated single cell sorting (FACS). Gating strategy for Tregs: (**b**–**d**) Tregs were analyzed by bulk RNA-seq (n = 3). A total of 1004 differentially expressed genes (DEGs) were identified by DESeq2 analysis. (**b**) Volcano plot of bulk RNA-seq (**c**) Heatmap of bulk RNA-seq (**d**) Gene set enrichment analysis (GSEA) of bulk analysis.

**Figure 2 ijms-26-06302-f002:**
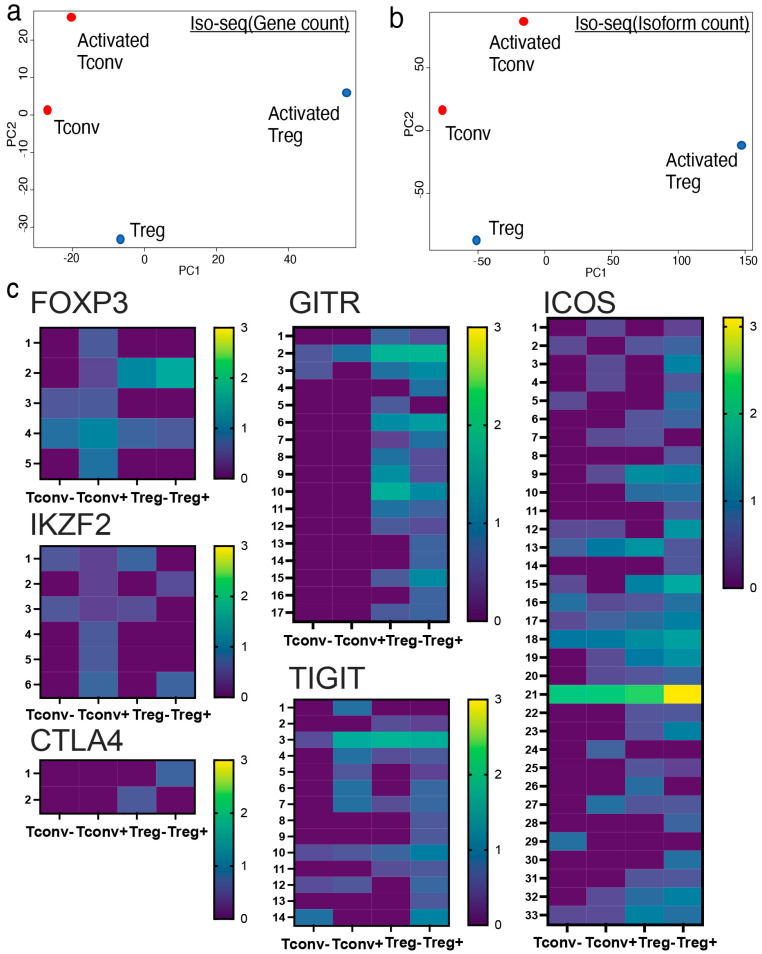
Iso-seq identified Treg-specific isoform usage upon activation. Isoform expression profiles of activated Tregs were analyzed by Iso-seq. (**a**,**b**) Principal component analysis (PCA) plot of Iso-seq shown by (**a**) total gene counts and (**b**) total isoform counts. (**c**) Isoform expression of Treg-related genes including FOXP3 (2 = full length, 4 = delta 2 isoform), IKZF-2 (3 = full length), CTLA-4 (2 = full length), GITR (2 = full length), TIGIT (3 = full length), and ICOS (21 = full length) shown by heatmap (log_10_TPM).

**Figure 3 ijms-26-06302-f003:**
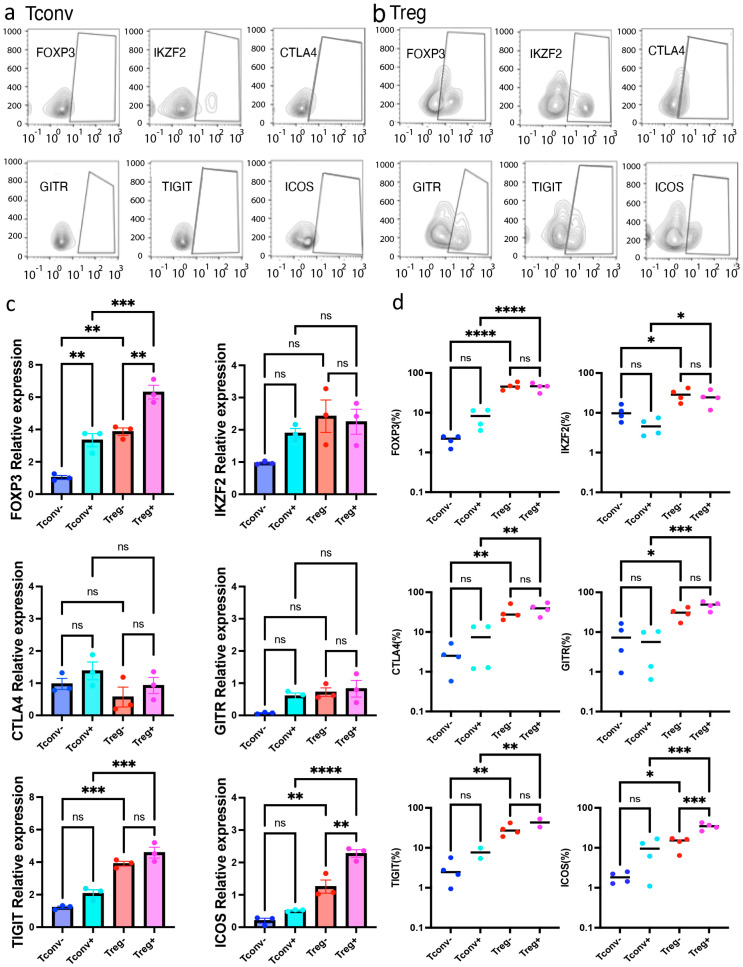
Activated Tregs, not resting Tregs, preferentially expressed ICOS. Tregs and Tconvs were analyzed by fluorescence activated single cell sorting (FACS) upon activation. (**a**) Representative FACS dot-plot of activated Tconvs upon TCR stimulation. (**b**) Representative FACS dot-plot of activated Tregs upon TCR stimulation. (**c**) qPCR validation of Treg-related genes identified by Iso-seq. Relative expression of target genes shown as relative expression (fold change) normalized by house-keeping genes (n = 3, Mean ± SEM). In Tregs, ICOS was significantly upregulated upon activation (*p* < 0.01). Results at *p*  <  0.001 were tested by ordinary one-way ANOVA and corrected for multiple comparisons using the Tukey adjustment. For the between-group analysis post-Tukey, stars were assigned as ** *p*  <  0.01, *** *p*  <  0.001, **** *p*  <  0.0001. (**d**) Expression of Treg-related molecules including FOXP3, IKZF-2, CTLA-4, GITR, TIGIT, and ICOS (n = 4, Mean ± SEM) measured by FACS. Results at *p*  <  0.001 were tested by ordinary one-way ANOVA and corrected for multiple comparisons using the Tukey adjustment. For the between-group analysis post-hoc Tukey test, stars were assigned as * *p*  <  0.05, ** *p*  <  0.01, *** *p*  <  0.001, **** *p*  <  0.0001.

**Figure 4 ijms-26-06302-f004:**
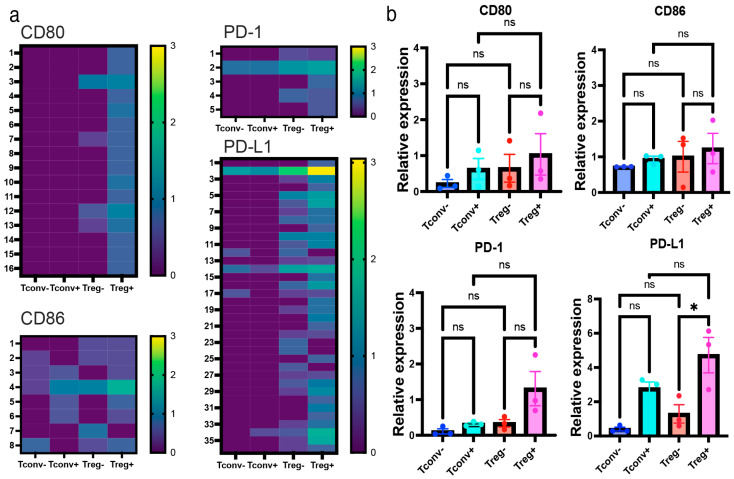
Iso-seq revealed Tregs preferentially expressed CD80, CD86, PD-1, and PD-L1. Isoform expression profile of activated Tregs analyzed by Iso-seq: (**a**) Isoform expression of CD80 (12 = full length), CD86 (4 = full length), PD-1 (2 = full length), and PD-L1 (2 = full length) shown by heatmap (log_10_TPM). (**b**) qPCR validation of the upregulation of CD80, CD86, PD-1, and PD-L1 in Tregs. Relative expression of target genes shown as relative expression (fold change) normalized by house-keeping genes (n = 3, Mean ± SEM). In Tregs, PD-L1 was significantly upregulated upon activation (*p* < 0.0001). Results at *p*  <  0.001 were tested by ordinary one-way ANOVA and corrected for multiple comparisons using the Tukey adjustment. For the between-group analysis post-hoc Tukey test, stars were assigned as * *p*  <  0.05.

**Figure 5 ijms-26-06302-f005:**
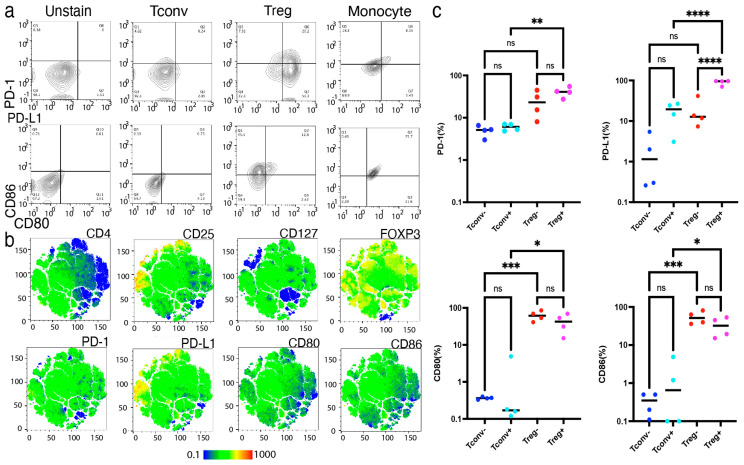
Activated Tregs, not resting Tregs, preferentially expressed PD-L1. Expression of CD80, CD86, PD-1, and PD-L1 in Tconvs and Tregs were analyzed by fluorescence activated single cell sorting (FACS) upon activation. (**a**) Representative FACS dot-plot of activated Tconvs and Tregs upon TCR stimulation. (**b**) Representative FACS t-SNE plot of Tregs upon activation. (**c**) Expression of CD80, CD86, PD-1, and PD-L1 measured by FACS (n = 4, Mean ± SEM). Results at *p*  <  0.001 were tested by ordinary one-way ANOVA and corrected for multiple comparisons using the Tukey adjustment. For the between-group analysis post-hoc Tukey test, stars were assigned as * *p*  <  0.05, ** *p*  <  0.01, *** *p*  <  0.001, **** *p*  <  0.0001.

**Figure 6 ijms-26-06302-f006:**
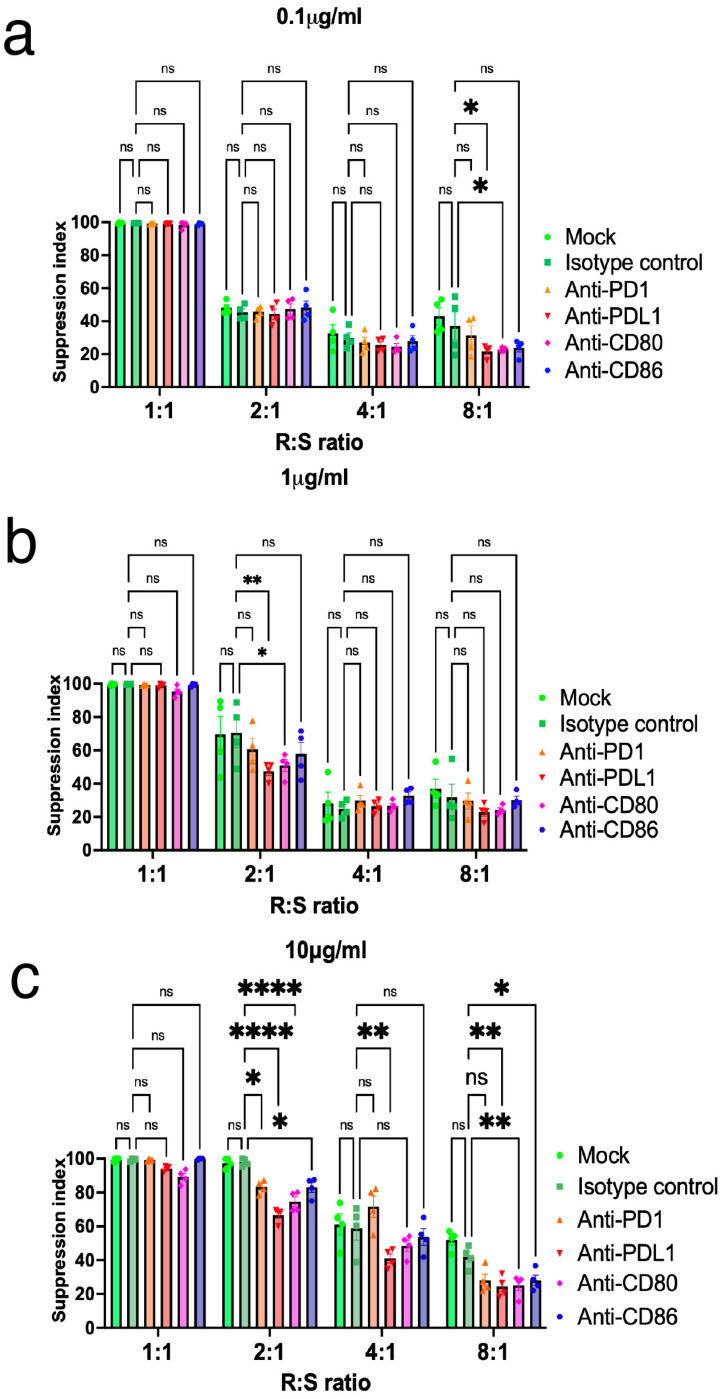
Inhibition of PD-L1 expression on Tregs results in the loss of suppressive function. Tregs suppression was blocked using CD80, CD86, PD-1, and PD-L1 antibodies. Tregs were pre-incubated with 0.1 μg/mL (**a**), 1 μg/mL (**b**), and 10 μg/mL (**c**) of each antibody before suppression assay. Proliferation of responder cells (Tconvs) was measured 96 h after poly clonal stimulation followed by co-culture with suppressor cells (Tregs). Results are expressed as Mean ± SEM (n = 4). Results at *p* <  0.001 were tested by ordinary one-way ANOVA and corrected for multiple comparisons using the Tukey adjustment. For the between-group analysis post-hoc Tukey test, stars were assigned as * *p*  <  0.05, ** *p*  <  0.01, **** *p*  <  0.0001.

## Data Availability

Data will be provided upon request.

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
