# Peer review of "Full-Length Transcriptome Sequencing Reveals Treg-Specific Isoform Expression upon Activation"

_ijms, 2025, doi:10.3390/ijms26136302_

Round 1
Reviewer 1 Report (New Reviewer)
Comments and Suggestions for Authors
Sato et.al investigate the differences in gene expression, isoform expression and protein expression on activated and resting Tregs and Teffs. They find that after activation, Tregs upregulate a distinct and unique isoform profile and propose methods to target these activated, and hence more suppressive Tregs, to improve cancer therapy.
While I found the concept and parts of the data interesting and of scientific importance, several aspects of this manuscript must be improved to ensure that the most can be extracted from the collected data and to thoroughly explore the broad scientific question the authors ask in their work.
Introduction: While the introduction offers a broad overview of the field as it relates to Tregs, isoform sequencing and what is known about the changes in gene expression after activation, I found it at times confusing to follow and has a flow that does not set up the scientific question the authors are asking. The introduction needs to be reworked to clearly set up what is known about Treg gene activation, the importance of different isoforms in Tregs before and after activation and the difference between Teffs and Tregs in this context and the clinical significance of the work the researchers are doing.
Methods
- -Were the PBMCs collected from healthy donors?
- Are all the Tregs actually Foxp3+? It states that you isolated for Foxp3+ Tregs but used a kit for CD4+CD25+CD127lo cell isolation, since sorting for Foxp3+ cannot be done as it is an intracellular transcription factor. I do not know whether it is truly accurate to call all the Tregs FOXP3+ as from the flow data showed later in the manuscript, not all Tregs are FOXP3+
- Were the cells sorted for FOXP3+ expression after isolation and only the FOXP3+ cells used as Tregs in the study? The term FACs is used, is this used for just referencing general flow cytometric analysis or for cell sorting to separate populations for further experiments?
- For the RNAseq, what cutoffs for log2FoldChange and adjusted P value were used to identify DEG? Did you adjust the P value for multiple comparisons?
- Why did you choose not to use CD8+ T cells in the suppression assay? The effects of Tregs on suppressing CD8+ function is critical in an anti-cancer immune response and it would be interesting to see whether there are any differences between the effects of Tregs on CD4+ or CD8+ T cells and how this is changed by the antibody blockade.
- Why were CD127+ decided upon as the effector CD4 comparative group. What CD4 subset is this predominantly made of?
Results
- This is a really useful dataset that both can enhance the analysis in this manuscript as well as be used by other researchers for their own studies. Therefore, more depth in the RNAseq analysis would really benefit the scientific merit of this article.
- What specific GSEA pathways were upregulated and downregulated between the two groups. Perhaps a graph showing all the GSEA pathways that were significantly different. What database did you run the GSEA on?
- Are there any other pathway analysis databases you can use to fully characterise the difference in gene expression between the two subsets?
Isoform sequencing
- Figure 2 and the results section related to this figure was a little confusing to follow. What are the y-axis for the heatmaps in figure 2c?
- Which data shows "Iso-seq identified that Tregs preferentially express the full length FOXP3 isoform, while Teffs expressed both isoforms almost equally". I found the presentation of the isoform sequencing results difficult to comprehend. This is a valuable resource and a large dataset that can really be explored and expanded upon.
Figure 3
- The figure could be made clearer to show that these are the representative flow plots of activated Tregs and Teffs. Since the results are focusing on the difference in activated and resting cells as well as between Tregs and Teffs, this figure could be adjusted to show representative plots of resting Teffs, activated Teffs, resting Tregs and activated Tregs to make the comparisons clearer.
- Error parts and different symbol shapes and/or colors for these bar graphs would made these data easier to interpret.
Figure 4 and expression of CD80 and PD-L1
- Why was expression of CD86 and PD-1 not possible to detect in Tregs, regardless of Iso-SEQ analysis?
- Similar to previous results, the presentation of the Iso-seq data can be improved with clearer axis labeling to further explain the data being shown.
- Why do you think CD86 and PD-1 were upregulated in the Iso-seq and not via the qPCR?
Figure 5 and PD-L1
- Again, the representative flow plots could be improved upon as mentioned above, along with the bar graphs.
- What is the basis for the conclusion that PD-L1 upregulation in activated Tregs may contribute to stable FOXP3 expression? The data shows that PD-L1 increases after activation, what is the link in this data to stable FOXP3?
Figure 6 and suppressive assays
- The data shows that the blockade of PD-L1 did not decrease FOXP3 expression in Tregs however one of your conclusions is that it is important for the maintenance of FOXP3 expression? The data shows that blocking PD-L1 decreases the suppressive ability of Tregs but the link to FOXP3 does not have data to support it. Does PD-L1 blockade downregulate genes that are induced by FOXP3 so even though FOXP3 itself isn't downregulated, the downstream effects are?
- The graphs for the suppression assay results can be improved to make clearer, since the small size of the symbols and the bars along with the lack of color makes identifying which group is which difficult, as well as which groups are significant from each other.
- It seems like anti-CD80 decreases the suppressive index, same as anti-CD86, in some experimental conditions. However the results discussion say that they didn't? This needs to be corrected.
Discussion
- The discussion overall is well written and puts the results into a bigger clinical context.
- However there are some inconsistencies between some parts of the results and the discussion
- Line 361: checkpoint blockade does not directly dampen foxp3 expression or the suppressive function of Tregs. However anti-PDL1 and to a lesser extend anti-CD80/CD86 do decrease the Treg suppressive activity. I think to be most correct, blocking any of these molecules on Tregs are a form of checkpoint blockade. The next line 362 says that PDL1 blockade does decrease Treg suppressive capacity.
- "Our data correspond with a previous description that the upregulation of CD80/CD86 occurred with PDL1 upregulation". The shown data does not show this directly. Are the cells that express PDL1 also CD80/CD86+. Are they all co-expressed before/after activation? The current data does not fully support this conclusion.
Author Response
Sato et.al investigate the differences in gene expression, isoform expression and protein expression on activated and resting Tregs and Teffs. They find that after activation, Tregs upregulate a distinct and unique isoform profile and propose methods to target these activated, and hence more suppressive Tregs, to improve cancer therapy.
While I found the concept and parts of the data interesting and of scientific importance, several aspects of this manuscript must be improved to ensure that the most can be extracted from the collected data and to thoroughly explore the broad scientific question the authors ask in their work.
(Response) Thank you for appreciation to our work. Please see our point-by-point responses.
Introduction: While the introduction offers a broad overview of the field as it relates to Tregs, isoform sequencing and what is known about the changes in gene expression after activation, I found it at times confusing to follow and has a flow that does not set up the scientific question the authors are asking. The introduction needs to be reworked to clearly set up what is known about Treg gene activation, the importance of different isoforms in Tregs before and after activation and the difference between Teffs and Tregs in this context and the clinical significance of the work the researchers are doing.
(Response) We carefully revised manuscript according to the suggestions.
Methods
- -Were the PBMCs collected from healthy donors?
(Response) Yes, they were purchased from the commercial vendor as written in the manuscript.
“Frozen human PBMCs isolated from healthy donors were purchased from Precision Medicine (n=4, for bulk RNA-seq) and Lonza (n=4, for other studies) were stored in liquid nitrogen.”
- Are all the Tregs actually Foxp3+? It states that you isolated for Foxp3+ Tregs but used a kit for CD4+CD25+CD127lo cell isolation, since sorting for Foxp3+ cannot be done as it is an intracellular transcription factor. I do not know whether it is truly accurate to call all the Tregs FOXP3+ as from the flow data showed later in the manuscript, not all Tregs are FOXP3+
(Response) Thank you for pointing this out. As the reviewer pointed out, it was not possible to isolate Treg according to the FOXP3 expression especially from human PBMCs. Moreover, human activated CD4+ T cells also slightly express low level of FOXP3 transiently. Therefore, it was not possible to conclude that all the sorted Tregs are FOXP3 positive and non-Tregs are FOXP3 negative.
- Were the cells sorted for FOXP3+ expression after isolation and only the FOXP3+ cells used as Tregs in the study? The term FACs is used, is this used for just referencing general flow cytometric analysis or for cell sorting to separate populations for further experiments?
(Response) To avoid confusion, we have modified terminology as suggested throughout the manuscript.
- For the RNAseq, what cutoffs for log2FoldChange and adjusted P value were used to identify DEG? Did you adjust the P value for multiple comparisons?
(Response) We provided cutoff values for adjusted P value.
“Differential gene expression was analyzed using DEseq2 (adjusted p-values < 0.01)”
- Why did you choose not to use CD8+ T cells in the suppression assay? The effects of Tregs on suppressing CD8+ function is critical in an anti-cancer immune response and it would be interesting to see whether there are any differences between the effects of Tregs on CD4+ or CD8+ T cells and how this is changed by the antibody blockade.
(Response) Thank you so much for your insightful suggestions. We previously used CD8+ T cells as a “responder cell” in suppression assay, however, due to the relatively smaller number of CD8+ T cells compared to CD4+ T cells, we finally used CD4+ T cells as a “responder cell” similarly to the other groups.
- Why were CD127+ decided upon as the effector CD4 comparative group. What CD4 subset is this predominantly made of?
(Response) We were not able to study CD4+ T cells in detail in the current study. They must contain naïve and effector T cells, however, it was not possible to provide the data on each subset such as Th1, Th2 and Th17.
Results
- This is a really useful dataset that both can enhance the analysis in this manuscript as well as be used by other researchers for their own studies. Therefore, more depth in the RNAseq analysis would really benefit the scientific merit of this article.
(Response) Thank you for your suggestion. We have added more in detailed RNA-seq analysis results in Supplementary Figure 2.
- What specific GSEA pathways were upregulated and downregulated between the two groups. Perhaps a graph showing all the GSEA pathways that were significantly different. What database did you run the GSEA on?
(Response) We used “C7: immunologic signature gene sets” in this study.
- Are there any other pathway analysis databases you can use to fully characterise the difference in gene expression between the two subsets?
(Response) We provided additional analysis results in Supplementary Figure 2.
Isoform sequencing
- Figure 2 and the results section related to this figure was a little confusing to follow. What are the y-axis for the heatmaps in figure 2c?
(Response) Y-axis showed different isoforms. We provided the caption for the clarity.
- Which data shows "Iso-seq identified that Tregs preferentially express the full length FOXP3 isoform, while Teffs expressed both isoforms almost equally". I found the presentation of the isoform sequencing results difficult to comprehend. This is a valuable resource and a large dataset that can really be explored and expanded upon.
(Response) We would like to apologize for the confusion. We provided figure caption.
Figure 3
- The figure could be made clearer to show that these are the representative flow plots of activated Tregs and Teffs. Since the results are focusing on the difference in activated and resting cells as well as between Tregs and Teffs, this figure could be adjusted to show representative plots of resting Teffs, activated Teffs, resting Tregs and activated Tregs to make the comparisons clearer.
(Response) Thank you so much for your insightful suggestions. We have modified figure captions.
- Error parts and different symbol shapes and/or colors for these bar graphs would made these data easier to interpret.
(Response) Thank you so much for your suggestions. We have modified symbol shapes and colors.
Figure 4 and expression of CD80 and PD-L1
- Why was expression of CD86 and PD-1 not possible to detect in Tregs, regardless of Iso-SEQ analysis?
(Response) CD86 and PD-1 was expressed in Treg at low level. The difference between qPCR/Iso-seq could be most likely due to the technical difference. While qPCR amplify short amplicon, Iso-seq count full length mRNA.
- Similar to previous results, the presentation of the Iso-seq data can be improved with clearer axis labeling to further explain the data being shown.
(Response) We provided explanation of the data.
- Why do you think CD86 and PD-1 were upregulated in the Iso-seq and not via the qPCR?
(Response) CD86 and PD-1 was expressed in Treg at low level. The difference between qPCR/Iso-seq is simply due to the technical difference. While qPCR amplify short amplicon, Iso-seq count full length mRNA.
Figure 5 and PD-L1
- Again, the representative flow plots could be improved upon as mentioned above, along with the bar graphs.
(Response) Thank you so much for your suggestions. We have modified figure captions.
- What is the basis for the conclusion that PD-L1 upregulation in activated Tregs may contribute to stable FOXP3 expression? The data shows that PD-L1 increases after activation, what is the link in this data to stable FOXP3?
(Response) As the reviewer pointed out, we did not have solid evidence that PD-L1 is directly induced by FOXP3.
Figure 6 and suppressive assays
- The data shows that the blockade of PD-L1 did not decrease FOXP3 expression in Tregs however one of your conclusions is that it is important for the maintenance of FOXP3 expression? The data shows that blocking PD-L1 decreases the suppressive ability of Tregs but the link to FOXP3 does not have data to support it. Does PD-L1 blockade downregulate genes that are induced by FOXP3 so even though FOXP3 itself isn't downregulated, the downstream effects are?
(Response) We apologize for the confusion. Similarly to previous response, we did show that PD-L1 blockade decrease suppressive function while FOXP3 expression was maintained. PD-L1 is suppressive molecule on Tregs and not directly interact with FOXP3.
- The graphs for the suppression assay results can be improved to make clearer, since the small size of the symbols and the bars along with the lack of color makes identifying which group is which difficult, as well as which groups are significant from each other.
(Response) Thank you for your valuable suggestions.
- It seems like anti-CD80 decreases the suppressive index, same as anti-CD86, in some experimental conditions. However, the results discussion say that they didn't? This needs to be corrected.
(Response) We modified the discussion as suggested.
“Nevertheless, CD80/CD86 blockade also partially inhibit T cell proliferation in the limited experimental conditions.”
Discussion
- The discussion overall is well written and puts the results into a bigger clinical context.
- However there are some inconsistencies between some parts of the results and the discussion
- Line 361: checkpoint blockade does not directly dampen foxp3 expression or the suppressive function of Tregs. However anti-PDL1 and to a lesser extend anti-CD80/CD86 do decrease the Treg suppressive activity. I think to be most correct, blocking any of these molecules on Tregs are a form of checkpoint blockade. The next line 362 says that PDL1 blockade does decrease Treg suppressive capacity.
(Response) We apologized for the confusion. We revised manuscript accordingly.
“Nevertheless, CD80/CD86 blockade also partially inhibit T cell proliferation in the limited experimental conditions.”
- "Our data correspond with a previous description that the upregulation of CD80/CD86 occurred with PDL1 upregulation". The shown data does not show this directly. Are the cells that express PDL1 also CD80/CD86+. Are they all co-expressed before/after activation? The current data does not fully support this conclusion.
(Response) We apologized for the confusion. We revised manuscript accordingly.
“Our findings do not contradict a previous description that the upregulation of CD80 and CD86 occurred with PD-L1 upregulation.”
Reviewer 2 Report (New Reviewer)
Comments and Suggestions for Authors
This paper demonstrates that FOXP3+ regulatory T cells (Tregs) regulate immune functions through specific isoforms, such as FOXP3 delta 2 and CTLA-4. Long-read sequencing (Iso-seq) revealed that Tregs possess a unique genome-wide isoform repertoire, including preferential expression of full-length FOXP3 and activation-dependent upregulation of ICOS and PD-L1. Although PD-L1 blockade did not affect FOXP3 expression, it significantly impaired the suppressive function of Tregs, suggesting that dynamic remodeling of the isoform repertoire upon T-cell receptor activation critically modulates Treg functionality. This study systematically characterizes the isoform landscape of Tregs, uncovering novel molecular mechanisms underlying their suppressive activity. The findings highlight PD-L1 and ICOS as potential therapeutic targets, providing a foundation for developing Treg-targeted immunotherapies in cancer and autoimmune diseases. The paper is well-designed and clearly written; however, some revisions have been marked in the PDF. Please refer to it.

Author Response
This paper demonstrates that FOXP3+ regulatory T cells (Tregs) regulate immune functions through specific isoforms, such as FOXP3 delta 2 and CTLA-4. Long-read sequencing (Iso-seq) revealed that Tregs possess a unique genome-wide isoform repertoire, including preferential expression of full-length FOXP3 and activation-dependent upregulation of ICOS and PD-L1. Although PD-L1 blockade did not affect FOXP3 expression, it significantly impaired the suppressive function of Tregs, suggesting that dynamic remodeling of the isoform repertoire upon T-cell receptor activation critically modulates Treg functionality. This study systematically characterizes the isoform landscape of Tregs, uncovering novel molecular mechanisms underlying their suppressive activity. The findings highlight PD-L1 and ICOS as potential therapeutic targets, providing a foundation for developing Treg-targeted immunotherapies in cancer and autoimmune diseases. The paper is well-designed and clearly written; however, some revisions have been marked in the PDF. Please refer to it.
(Response) Thank you so much for insightful comments. We revised the manuscript accordingly.
This manuscript is a resubmission of an earlier submission. The following is a list of the peer review reports and author responses from that submission.
Round 1
Reviewer 1 Report
Comments and Suggestions for Authors
You did not address almost all the comments given previously.
2 Full-Length Transcriptome Sequencing Reveals Treg-Specific Isoform Expression Upon Activation
Did you study the expression of the protein isoforms https://www.ncbi.nlm.nih.gov/mesh/68020033 or the RNA spliced variants? Not all RNA spliced variants encode proteins. Who has invented this method and named it Iso-seq?
17 Why do you use different gene symbols: PD-L1 and PDL1, CTLA4 and CTLA-4, PD-1 and PD1? It is scientifically accurate to go with unique official gene symbols and names (https://www.genenames.org/) like Applied Biosystems Taq-Man assays
https://www.thermofisher.com/taqman-gene-expression/product/Hs00204257_m1 CD274
https://www.thermofisher.com/taqman-gene-expression/product/Hs01550088_m1 PDCD1
Which gene symbols are utilized by the RNA-seq and Iso-sec softwares? I want to see the original files with the list of DEGs (RNA-seq) and corresponding spliced variants (Iso-sec).
Also, I need to know the accession numbers for all spliced variants Fig. 2C, 4A and Supplementary Figure 2.
Did you discover the novel spliced variants?
80 You must provide the catalogue numbers along with the company name for all Materials and Methods.
81 It is really the PBMC isolation protocol? This part is not reproducible by others. Provide all necessary parameters – volume/quantity, time, temperature etc.
82 Is a correct name Precision for Medicine?
You must provide in section 2.1 all information given by the company about “three independent PBMC donors used in this study” (Line 165).
What can you say about the forth sample used in FACS (Line 255) and really 4 dots on Fig 4D?
86 How were CD4+ T cells isolated?
87 Serum or plasma? Why was serum derived from plasma, whereas serum is usually obtained directly from blood? Was the plasma-derived serum used for some reasons?
108 A260/280 ratio >2.0 and/or RIN>7.5 were set as cut-off values for further analysis.
In previous version of this manuscript, you specified the cut-off value <2.0 that was totally incorrect. The lower the A260/280 value, the less pure the RNA. However, the A260/280 ratio >2.0 is also doubtful. This study is utterly dependent on the RNA quality. More than before, I have to see the original Excel files with the RNA concentration, A260/280, A230/260 and RIN values, as well as an accurate record of which experiments each RNA sample was used in.
108, 123, 134 Before RIN values were different >7.5 and >7.0. Was it a typo or reality?
111 Your RT-qPCR experiments are not possible to replicate by others. You should provide all details explicitly. It is recommended to follow the MIQE guidelines https://pubmed.ncbi.nlm.nih.gov/19246619/
https://pubmed.ncbi.nlm.nih.gov/38290180/
Significant variation in expression levels (Figures 3C and 4B) indicates that your real-time RT-qPCR method is not robust and could produce wrong results. Conditionally identical samples must produce similar RT-qPCR results. Especially, when your data show that both naïve and activated cells have identical phenotypes (Supplementary Figure 1)
To be sure, I need from you next data: Excel files extracted from the original SDS files with sample names (including positive control, no template control, no RT control), cDNA input, threshold and Ct values, the ΔCT equation and final results shown on Figures 3C and 4B.
Why did not you validate the most important Iso-sec findings by using specific Taq-Man assays?
223 (log? TPM) What is a base?
257 Rephrase next: For between group analysis post-Tukey
Author Response
*Please see uploaded attachment to see the images used in the response (IJMS Reviwer1.pdf). I was not able to uploaded the images in the response. *
2 Full-Length Transcriptome Sequencing Reveals Treg-Specific Isoform Expression Upon Activation
Did you study the expression of the protein isoforms https://www.ncbi.nlm.nih.gov/mesh/68020033 or the RNA spliced variants? Not all RNA spliced variants encode proteins. Who has invented this method and named it Iso-seq?
(Response) Thank you so much for asking on the isoform expression which is the main subject of our study. “Iso-Seq” is sequencing methods for full-length cDNA using PacBio SMRT sequencing technology. As you can see supplementary table, not all the isoforms are protein-coding, but I believe it is informative to include non-coding splice variants. Also, we have identified several “de novo” transcripts which was not previously reported, however, due to the novelty of this technology, it may possible to identify novel minor de novo isoforms which has been (or will be) confirmed by other Iso-seq results. To avoid confusion, we did not specify de novo transcripts and non-coding transcript in the manuscript. Honestly, the data is little bit beyond current analytical pipelines which are usually optimized for short-read sequencing, not for long-read/ full length sequencing.
Iso-Seq (PacBio)
https://www.pacb.com/products-and-services/applications/rna-sequencing/
17 Why do you use different gene symbols: PD-L1 and PDL1, CTLA4 and CTLA-4, PD-1 and PD1? It is scientifically accurate to go with unique official gene symbols and names (https://www.genenames.org/) like Applied Biosystems Taq-Man assays
https://www.thermofisher.com/taqman-gene-expression/product/Hs00204257_m1 CD274
https://www.thermofisher.com/taqman-gene-expression/product/Hs01550088_m1 PDCD1
(Response) I would like to apologize for the different gene symbols. We have uniformed the terminology through the manuscript.
Which gene symbols are utilized by the RNA-seq and Iso-sec softwares? I want to see the original files with the list of DEGs (RNA-seq) and corresponding spliced variants (Iso-sec).
Also, I need to know the accession numbers for all spliced variants Fig. 2C, 4A and Supplementary Figure 2.
Did you discover the novel spliced variants?
(Response) Please see our previous response. We have identified several “de novo” transcripts which was not previously reported, however, due to the novelty of this technology, it may possible to identify novel minor de novo isoforms which has been (or will be) confirmed by other Iso-seq results. To avoid confusion, we did not specify de novo transcripts and non-coding transcript in the manuscript.
Also, thank you so much for asking the correlation between RNA-seq/Iso-seq. We are not able to see full correlation between RNA-seq/Iso-seq while the gene expressions were comparable. Because RNA was processed and sequenced by totally different methods, it may not be fully identical in nature. It was commonly believed that short-read is optimal for gene expression profile while long-read is optimal for identifying minor isoform which was not precisely counted by short-read sequencing. Due to high sequencing demands, Iso-seq is still very expensive and the experiments are not easily repeated by limited research budget. Limited sample pooling capacity makes it difficult to become common laboratory assays. However, it may possible to become more accessible if the technologies become more popular. Therefore, we would like to publish the paper on Iso-seq.
80 You must provide the catalogue numbers along with the company name for all Materials and Methods.
(Response) Thank you so much for pointing this out. Now, we have provided the manufacture’s information together with catalogue numbers for all the reagents used in this study.
81 It is really the PBMC isolation protocol? This part is not reproducible by others. Provide all necessary parameters – volume/quantity, time, temperature etc.
(Response) Thank you so much for suggestion. As the reviewer suggested, this is not PBMC isolation protocol because PBMC has been isolated by commercial vendor. We have modified title and provided more detailed information as suggested.
82 Is a correct name Precision for Medicine?
You must provide in section 2.1 all information given by the company about “three independent PBMC donors used in this study” (Line 165).
What can you say about the forth sample used in FACS (Line 255) and really 4 dots on Fig 4D?
(Response) We would like to apologize for the confusion. Precision for Medicine, Inc. is the correct company name. To avoid confusion, we provided full commercial name of the company. For bulk RNA-seq, we used three independent PBMC donors purchased from Precision for Medicine. For other studies including qPCR, we used four independent PBMC donors purchased from Lonza. We clarified the source of PBMCs in the method section.
86 How were CD4+ T cells isolated?
(Response) We have provided the isolation in the methods section.
87 Serum or plasma? Why was serum derived from plasma, whereas serum is usually obtained directly from blood? Was the plasma-derived serum used for some reasons?
(Response) Human serum is a common research material used in the human cell culture.
https://www.thermofisher.com/jp/en/home/references/protocols/cell-and-tissue-analysis/elisa-protocol/elisa-sample-preparation-protocols/plasma-and-serum-preparation.html#:~:text=After%20collection%20of%20the%20whole,resulting%20supernatant%20is%20designated%20serum.
As far as I know, plasma was collected from blood type AB donors and clots were removed from the plasma and the remaining blood products were considered as “plasma” after removal of clotting factors.
108 A260/280 ratio >2.0 and/or RIN>7.5 were set as cut-off values for further analysis.
In previous version of this manuscript, you specified the cut-off value <2.0 that was totally incorrect. The lower the A260/280 value, the less pure the RNA. However, the A260/280 ratio >2.0 is also doubtful. This study is utterly dependent on the RNA quality. More than before, I have to see the original Excel files with the RNA concentration, A260/280, A230/260 and RIN values, as well as an accurate record of which experiments each RNA sample was used in.
(Response) For the clarity we have provided all the raw values for RNA quality in the following section.
108, 123, 134 Before RIN values were different >7.5 and >7.0. Was it a typo or reality?
(Response) Indeed, we used both RIN cut-off values either >7.5 and >7.0 according to the sample condition. To avoid confusion, we uniformly proposed RIN 7.5, however, all the sample showed RIN >8.0 except one sample which has RIN 6.8. We provided the statement in the method section. We have pictures of each RNA analysis using Bioanalyzer.
111 Your RT-qPCR experiments are not possible to replicate by others. You should provide all details explicitly. It is recommended to follow the MIQE guidelines https://pubmed.ncbi.nlm.nih.gov/19246619/
https://pubmed.ncbi.nlm.nih.gov/38290180/
Significant variation in expression levels (Figures 3C and 4B) indicates that your real-time RT-qPCR method is not robust and could produce wrong results. Conditionally identical samples must produce similar RT-qPCR results. Especially, when your data show that both naïve and activated cells have identical phenotypes (Supplementary Figure 1)
(Response) We would like to apologize for the quality of RT-qPCR assays. Since we are analyzing different donors, it is possible to observe donor-to-donor variance. Similarly to qPCR, we have observed donor-to-donor variance in flow cytometry. Ideally, we have to repeat experiments to consolidate the data, however, it was not readily possible at this time moments due to several limitations.
To be sure, I need from you next data: Excel files extracted from the original SDS files with sample names (including positive control, no template control, no RT control), cDNA input, threshold and Ct values, the ΔCT equation and final results shown on Figures 3C and 4B.
(Response) We would like to apologize for the quality of RT-qPCR assays and provided the sample raw values for the assay validation. We performed qPCR just for the validation of RNA-seq, Iso-seq and Flow Cytometry. However, we also admit the quality of qPCR and have provided the limitation in the discussion as follows.
“ Moreover, due to the limited sample number, we were not able to perform qPCR for multiple donors (n>10) to consolidate our findings in RNA-seq, Iso-seq and Flow cytometry. Follow up study is warranted to consolidate isoform expression with regards to each key gene related to Treg function including FOXP3, IKZF-2 and PD-L1.”
Why did not you validate the most important Iso-sec findings by using specific Taq-Man assays?
(Response) It was not possible to design isoform specific Taq-man assays especially with FOXP3. Also, it was not possible to cover all isoform specific Taq-man assays, but we understand the importance of validation with regards to each isoform related to Treg functionality.
223 (log? TPM) What is a base?
(Response) We would like to apologize. Log10 (TPM)
257 Rephrase next: For between group analysis post-Tukey
(Response) We have modified the sentence. “For between group analysis post-hoc Tukey test, ”

Reviewer 2 Report
Comments and Suggestions for Authors
In the manuscript entitled “Full-length transcriptome sequencing reveals Treg-specific isoform expression upon activation”, the authors report original findings supporting a specific role for FOXP3 full-length (FL) isoform in activated regulatory T cells via adoption of long reads RNA sequencing in their experimental design.
In particular, using relative quantitative transcripts expression as well as via parallel isoform-species detection, they demonstrate (a) that activated Tregs preferentially express FL-FOXP3 rather than the commonly expressed Delta2 (Ex2-) variant, and (b) that a specific pattern of immune checkpoint regulating molecules expression (including FOXP3, CTLA-4 and IKZF2) can be found in Tregs compared to Teffs, some of which with actionable value (such as the ICOS and PD-L1 co-expression in activated Tregs).
Finally, using appropriate experimental design and sufficient controls, they demonstrate that PD-L1 blockade in activated Tregs does not affect FOXP3 expression despite significantly reducing Tregs suppressive function.
Overall, the study conveys original and innovative findings advancing current knowledge of regulatory T cells biology along with supporting the paradigm disruptive concept that the expression of established immune checkpoint regulating molecules such as PD-L1 are not restricted to antigen presenting cells but that they play a (potentially actionable) role also in regulated T cells under specific circumstances (namely, upon activation) worth being further explored and pharmacologically/clinically weighted.
For this reason, this reviewer consider the manuscript worth of publication on this journal.
Nonetheless, minor key problems apply to some of the figure composites which result of difficult if not unreadable level along with minor stylistic problems challenging the reader comprehension as reported below.
Such minor problems need to be addressed by the authors for the manuscript to be accepted by this reviewer. Specifically,
Figure 1d -> Unreadable X and Y axis wording. Need to separate the graphs from any currently applied number and letter and re-apply larger but less simplified indicative readable values/words to the same graphs-> eg by reducing the numbering on the Y axis ([0 – 0.2 – 0.4] [0 - 0.5 – 1.0] and the X axis (0- 10,000 – 20,000) so to be able to enlarge them to a minimum readable size.
Fig 2a-b -> numbering on the X/Y axis is NOT legible. increase the font size on graph (eg bring number size to the same level of the letters in the figure)
Similarly, X/Y axis letters and numbers on Figure 3a-b cannot be appreciated – suggest to scanning resolution and highlight a few refs values while leaving intermediate as tags (eg increase font size on 200 – 400 - 600 – 800 – 1000 while leaving intermediate values [100 – 300 – 500 – 700 – 900] as simple dashes on the Y axis)
Identical key problem applies to figure composite 6 in which the wording in the column graph is negligibly readable due to excessive small letters size. It is advisable to enlarge font characters of the column graphs and in minor level the associated column graphs by reducing in parallel the area occupied by the heat color graphs on their right by at least 1/5th of the current space (as they have done already for figure composite 4). This should guarantee a minimum satisfactory understanding of the figure by the reader without need to zoom in manually on the composite.
In the text, under lanes 309 and 310 where the repeated wording “in contrast” could be replaced with “on the other hand” (or the like)
Under Discussion at the end of the period spanning from lane 349 to 352, the authors end the period sentence with the wording “for the same” without adding further object (same what?). Upon reading the full period, this reviewer finds such wording either unnecessary (the sentence has complete meaning without it), or incomplete (in the latter case they’d need to complete the sentence or explain it further with a new added sentence). Either way, this wording needs to be either removed or completed.
Upon such minor but significant corrections in the manuscript, this reviewer finds the manuscript of publication level and both of general (immuno-biology) and specific (molecular medicine) interest for the international scientific community involved in life-science and translational research.
Author Response
In the manuscript entitled “Full-length transcriptome sequencing reveals Treg-specific isoform expression upon activation”, the authors report original findings supporting a specific role for FOXP3 full-length (FL) isoform in activated regulatory T cells via adoption of long reads RNA sequencing in their experimental design.
In particular, using relative quantitative transcripts expression as well as via parallel isoform-species detection, they demonstrate (a) that activated Tregs preferentially express FL-FOXP3 rather than the commonly expressed Delta2 (Ex2-) variant, and (b) that a specific pattern of immune checkpoint regulating molecules expression (including FOXP3, CTLA-4 and IKZF2) can be found in Tregs compared to Teffs, some of which with actionable value (such as the ICOS and PD-L1 co-expression in activated Tregs).
Finally, using appropriate experimental design and sufficient controls, they demonstrate that PD-L1 blockade in activated Tregs does not affect FOXP3 expression despite significantly reducing Tregs suppressive function.
Overall, the study conveys original and innovative findings advancing current knowledge of regulatory T cells biology along with supporting the paradigm disruptive concept that the expression of established immune checkpoint regulating molecules such as PD-L1 are not restricted to antigen presenting cells but that they play a (potentially actionable) role also in regulated T cells under specific circumstances (namely, upon activation) worth being further explored and pharmacologically/clinically weighted.
For this reason, this reviewer consider the manuscript worth of publication on this journal.
Nonetheless, minor key problems apply to some of the figure composites which result of difficult if not unreadable level along with minor stylistic problems challenging the reader comprehension as reported below.
(Response) Thank you so much for appreciation of our study. We sincerely appreciate insightful and thoughtful comments from the reviewer. We fully agreed with the revision requests and please see our point-by-point response below.
Such minor problems need to be addressed by the authors for the manuscript to be accepted by this reviewer. Specifically,
Figure 1d -> Unreadable X and Y axis wording. Need to separate the graphs from any currently applied number and letter and re-apply larger but less simplified indicative readable values/words to the same graphs-> eg by reducing the numbering on the Y axis ([0 – 0.2 – 0.4] [0 - 0.5 – 1.0] and the X axis (0- 10,000 – 20,000) so to be able to enlarge them to a minimum readable size.
(Response) Thank you so much for pointing this out. We have modified figure captions accordingly.
Fig 2a-b -> numbering on the X/Y axis is NOT legible. increase the font size on graph (eg bring number size to the same level of the letters in the figure)
(Response) Thank you so much for pointing this out. We have modified figure captions accordingly.
Similarly, X/Y axis letters and numbers on Figure 3a-b cannot be appreciated – suggest to scanning resolution and highlight a few refs values while leaving intermediate as tags (eg increase font size on 200 – 400 - 600 – 800 – 1000 while leaving intermediate values [100 – 300 – 500 – 700 – 900] as simple dashes on the Y axis)
(Response) Thank you so much for pointing this out. We have modified figure captions accordingly.
Identical key problem applies to figure composite 6 in which the wording in the column graph is negligibly readable due to excessive small letters size. It is advisable to enlarge font characters of the column graphs and in minor level the associated column graphs by reducing in parallel the area occupied by the heat color graphs on their right by at least 1/5th of the current space (as they have done already for figure composite 4). This should guarantee a minimum satisfactory understanding of the figure by the reader without need to zoom in manually on the composite.
(Response) Thank you so much for pointing this out. We have modified the figure accordingly.
In the text, under lanes 309 and 310 where the repeated wording “in contrast” could be replaced with “on the other hand” (or the like)
(Response) Thank you so much for pointing this out. We have modified the text accordingly.
Under Discussion at the end of the period spanning from lane 349 to 352, the authors end the period sentence with the wording “for the same” without adding further object (same what?).
(Response) Thank you so much for pointing this out. We have modified the text accordingly.
Upon reading the full period, this reviewer finds such wording either unnecessary (the sentence has complete meaning without it), or incomplete (in the latter case they’d need to complete the sentence or explain it further with a new added sentence). Either way, this wording needs to be either removed or completed.
(Response) We would like to apologize for the confusion. We carefully corrected the manuscript entirely.
Upon such minor but significant corrections in the manuscript, this reviewer finds the manuscript of publication level and both of general (immuno-biology) and specific (molecular medicine) interest for the international scientific community involved in life-science and translational research.
(Response) Again, we all sincerely appreciate thoughtful comments from the reviewer. We believe our revision according to the comments improved the quality of the manuscript.
Round 2
Reviewer 1 Report
Comments and Suggestions for Authors
This study has serious flaws, more experiments are needed, and the study was not conducted properly.
Iso-Seq was invented in 2015 and to date 351 relevant studies have been published. https://pubmed.ncbi.nlm.nih.gov/?term=Iso-Seq&sort=date&ac=yes The dynamics of publications shows that Iso-Seq as a relatively new method still remains uncertain and requires verification by traditional methods such as RNA-Seq and real-time RT-qPCR. All three methods provide both sequencing and discrimination of different RNA splice variants and corresponding expression data.
The authors acknowledged in their response to my comments that We are not able to see full correlation between RNA-seq/Iso-seq while the gene expressions were comparable.
It is normal that different sequencing approaches do not provide 100% correlation. However, the raw RNA-Seq data (Supplementary Tables) do not show all the genes whose expression was specifically examined in this study (Figures 3C and 4B), and the Iso-Seq table does not show gene symbols. Thus, it is not possible to understand how the RNA-Seq and Iso-Seq data correlate.
Real-time RT-qPCR is a classic reference method for RNA sequencing. Real-time RT-qPCR can accurately quantify expression levels and distinguish splice variants if used correctly. Real-time RT-qPCR raw data revealed that gene expression results (Figures 3C and 4B) are completely invalid.
The HPRT gene, as a housekeeping gene, should have constant expression under all experimental conditions. HPRT expression levels changed DIFFERENTLY following Teff cell activation, with Teff1 and 2 samples showing 6.05- and 8.43-fold decreases, while Teff3 showed an 11.38-fold increase. Thus, HPRT is not a valid reference gene for Teff activation experiments. Moreover, such a huge discrepancy indicates a complete sample inconsistency, not what the authors believe. Since we are analyzing different donors, it is possible to observe donor-to-donor variance. Similarly to qPCR, we have observed donor-to-donor variance in flow cytometry.
In addition, the RT-qPCR file shows many serious errors:
- Standard Curve, but it is Relative ddCt (see Materials and Methods).
- Different TaqMan assays should be analysed individually and not all together.
- The Ct values of the housekeeping gene should be significantly lower than the Ct values of the genes being studied.
- Baseline End cycle must be the same for the sample array and the specific TaqMan assay.
- To eliminate well and sample variations, multiplex assays using FAM and VIC reporter dyes for the gene of interest and the reference gene are highly recommended.
- All mandatory controls, including positive, no template and no RT, are absent.
- Authors did not answer my questions about the cDNA input, the ddCt equation on and the final results shown on Figures 3C and 4B.
The authors also ignored my request to show raw data on RNA concentration and A260/280 and A230/260 ratios.
The FACS experiments are not related to gene expression data, since completely different blood samples were used. Moreover, as the authors acknowledged Similarly to qPCR, we have observed donor-to-donor variance in flow cytometry.
In summary, the authors found new spliced variants but chose not to disclose them, thereby reducing the novelty of the study. RNA-Seq and Iso-Seq data are not comparable. Real-time RT-PCR is completely wrong and cannot confirm even overall gene expression. Enormous sample and/or donor variation. FACS cannot confirm gene expression data observed across different sample/donor arrays from different companies.
Author Response
This study has serious flaws, more experiments are needed, and the study was not conducted properly.
Iso-Seq was invented in 2015 and to date 351 relevant studies have been published. https://pubmed.ncbi.nlm.nih.gov/?term=Iso-Seq&sort=date&ac=yes The dynamics of publications shows that Iso-Seq as a relatively new method still remains uncertain and requires verification by traditional methods such as RNA-Seq and real-time RT-qPCR. All three methods provide both sequencing and discrimination of different RNA splice variants and corresponding expression data.
(Response) Thank you so much for the thoughtful and insightful comments. We fully agreed with the reviewer. Iso-seq is not very new methods, however, the results should be carefully interpreted together with the RNA-seq and qPCR data. Therefore, we have included RNA-seq and qPCR analysis of Tregs together with Iso-seq analysis.
The authors acknowledged in their response to my comments that We are not able to see full correlation between RNA-seq/Iso-seq while the gene expressions were comparable.
It is normal that different sequencing approaches do not provide 100% correlation. However, the raw RNA-Seq data (Supplementary Tables) do not show all the genes whose expression was specifically examined in this study (Figures 3C and 4B), and the Iso-Seq table does not show gene symbols. Thus, it is not possible to understand how the RNA-Seq and Iso-Seq data correlate.
(Response) We are sorry, but we have provided the expressions of the Treg related genes in the supplementary table showing RNA-seq results from the initial submission by the gene names. Now, we have provided gene names of 140,000 isoforms detected by Iso-seq accordingly, initially shown by ENSEMBL Gene ID. However, as we discussed in the past, gene expression profile may not be fully identical among the different approaches.
Real-time RT-qPCR is a classic reference method for RNA sequencing. Real-time RT-qPCR can accurately quantify expression levels and distinguish splice variants if used correctly. Real-time RT-qPCR raw data revealed that gene expression results (Figures 3C and 4B) are completely invalid.
(Response) Thank you for the suggestions. Hypothetically, it may possible to distinguish isoforms by Taqman assay, however, not all the assays are designed to distinguish isoforms. In most cases, the primers were designed to detect all isoforms (not the specific isoform variants) especially in the gene expression assay. We agreed that our qPCR is not optimal and needs to be improved. Therefore, it would be possible to remove Figures 3C and 4B and move them to supplementary figures as suggested. I would like to ask editor’s opinion.
The HPRT gene, as a housekeeping gene, should have constant expression under all experimental conditions. HPRT expression levels changed DIFFERENTLY following Teff cell activation, with Teff1 and 2 samples showing 6.05- and 8.43-fold decreases, while Teff3 showed an 11.38-fold increase. Thus, HPRT is not a valid reference gene for Teff activation experiments.
(Response) In general, HPRT is universally used as internal control in the real-time qPCR assays. I did understand that the qPCR condition is not optimal, however, it may be possible to validate RNA-seq/Iso-seq experiments together with flow cytometry as we insisted.
Moreover, such a huge discrepancy indicates a complete sample inconsistency, not what the authors believe. Since we are analyzing different donors, it is possible to observe donor-to-donor variance. Similarly to qPCR, we have observed donor-to-donor variance in flow cytometry.
In addition, the RT-qPCR file shows many serious errors:
- Standard Curve, but it is Relative ddCt (see Materials and Methods).
- Different TaqMan assays should be analysed individually and not all together.
- The Ct values of the housekeeping gene should be significantly lower than the Ct values of the genes being studied.
- Baseline End cycle must be the same for the sample array and the specific TaqMan assay.
- To eliminate well and sample variations, multiplex assays using FAM and VIC reporter dyes for the gene of interest and the reference gene are highly recommended.
- All mandatory controls, including positive, no template and no RT, are absent.
- Authors did not answer my questions about the cDNA input, the ddCt equation on and the final results shown on Figures 3C and 4B.
(Response) Thank you so much for the technical suggestions. However, there is no point to discuss on the experimental detail not included even in the supplementary materials. Taq-man gene expression assay is robust and reproducible methods compared to the other methods. Moreover, our qPCR results are compatible with the other assays including RNA-seq/ Flow cytometry. Therefore, we may conclude that we are able to achieve similar results across different modalities.
The authors also ignored my request to show raw data on RNA concentration and A260/280 and A230/260 ratios.
(Response) In the previous discussions, we have provided all the data measured by bioanalyzer which are more specific and sensitive compared to the nanodrop (A260/280). We used Nanodrop measurements as general screening methods and the RNA quality is further validated by bioanalyzer.
The FACS experiments are not related to gene expression data, since completely different blood samples were used. Moreover, as the authors acknowledged Similarly to qPCR, we have observed donor-to-donor variance in flow cytometry.
(Response) We believed that the confirmation of protein expressions can support the gene expression data.
In summary, the authors found new spliced variants but chose not to disclose them, thereby reducing the novelty of the study. RNA-Seq and Iso-Seq data are not comparable. Real-time RT-PCR is completely wrong and cannot confirm even overall gene expression. Enormous sample and/or donor variation. FACS cannot confirm gene expression data observed across different sample/donor arrays from different companies.
(Response) We are sorry for the confusion. We have shown all the de novo isoform in the supplementary table from the initial submissions. However, it may not possible to validate >3000 de novo transcripts in this study. Therefore, we would like to clarify that we are open to distribute the data.
Round 3
Reviewer 1 Report
Comments and Suggestions for Authors
This study has flaws in its overall design, and my comments addressed these issues, as well as the fact that methodological problems with real-time RT-qPCR completely ruined the validation of the Iso-Seq data. It is absolutely wrong to test different samples from different donor cohorts with different methods. As a result, you cannot compare datasets obtained by different assays, nor validate the Iso-Seq results using RT-PCR and FACS. Your last responses are a clear display of the confirmation bias https://en.wikipedia.org/wiki/Confirmation_bias
The table with RNA samples, RIN values ​​and methods used (author_response.pdf, review round 1) revealed a complete inconsistency in the experimental flow and, moreover, contradicts the manuscript. Why do you show RT-qPCR data for only 3 donors in the manuscript (Figs. 3C and 4B), while in the table the 4 donors are numbered as 5, 6, 7 and 8? Moreover, the RNA-Seq tests used a separate set of donors (1, 2 and 3), which makes the comparison between RNA-Seq and Iso-Seq completely invalid. It is wholly unclear how many donors were used and which RNA sample was obtained from which donor blood sample and from which company, which greatly complicates the peer review process.
In addition, the Iso-Seq Expression (Count) and Normalized Expression data in Supplementary Table Sheet 4 are not annotated according to the RNA samples (Treg5, Activated Treg5, Teff5, and Activated Teff5) used in the table in the author_response.pdf.
This study lacks scientific merit because it abuses basic scientific methodology - new knowledge must be verified by different analyses and can be reproduced by others.